# Targeting Measurable Residual Disease (MRD) in Acute Myeloid Leukemia (AML): Moving beyond Prognostication

**DOI:** 10.3390/ijms24054790

**Published:** 2023-03-01

**Authors:** Ing S. Tiong, Sun Loo

**Affiliations:** 1Peter MacCallum Cancer Centre, Melbourne, VIC 3000, Australia; 2The Alfred Hospital, Melbourne, VIC 3004, Australia; 3Australian Centre for Blood Diseases, Monash University, Melbourne, VIC 3004, Australia; 4Walter and Eliza Hall Institute of Medical Research, Parkville, VIC 3052, Australia; 5The Northern Hospital, Epping, VIC 3076, Australia

**Keywords:** AML, MRD, relapse, novel therapy, *NPM1*

## Abstract

Measurable residual disease (MRD) assessment in acute myeloid leukemia (AML) has an established role in disease prognostication, particularly in guiding decisions for hematopoietic cell transplantation in first remission. Serial MRD assessment is now routinely recommended in the evaluation of treatment response and monitoring in AML by the European LeukemiaNet. The key question remains, however, if MRD in AML is clinically actionable or “does MRD merely portend fate”? With a series of new drug approvals since 2017, we now have more targeted and less toxic therapeutic options for the potential application of MRD-directed therapy. Recent approval of *NPM1* MRD as a regulatory endpoint is also foreseen to drastically transform the clinical trial landscape such as biomarker-driven adaptive design. In this article, we will review (1) the emerging molecular MRD markers (such as non-*DTA* mutations, *IDH1/2*, and *FLT3*-ITD); (2) the impact of novel therapeutics on MRD endpoints; and (3) how MRD might be used as a predictive biomarker to guide therapy in AML beyond its prognostic role, which is the focus of two large collaborative trials: AMLM26 INTERCEPT (ACTRN12621000439842) and MyeloMATCH (NCT05564390).

## 1. Introduction

Although intensive chemotherapy induces complete remission (CR) in approximately 70% of AML cases, 30–80% relapse within the first 2 years, representing a major barrier to long-term cure in AML [1]. The assessment of AML disease response status has been traditionally based on a 5% threshold after morphologic assessment of ~200–500 cells under the microscope; however, leukemia cells could remain at 1 in >10^6^. The assessment of subclinical levels of leukemia, namely measurable residual disease (MRD), provides an integrated assessment beyond baseline characteristics, such as pharmacokinetic resistance, therapeutic sensitivity, immune microenvironment, and other patient and external factors. MRD assessment is now routinely recommended in the evaluation of treatment response and monitoring in AML by the European LeukemiaNet (ELN), using multiparameter flow cytometry MRD (MFC-MRD) based on the combination of diagnostic leukemia-associated immunophenotype (LAIP) and different from normal (DfN) immunophenotype, and validated molecular markers, namely *NPM1*, *RUNX1::RUNX1T1*, *CBFB::MYH11*, and *PML::RARA* [2,3,4]. Notably, *NPM1* MRD was recently approved as a regulatory approval endpoint in a clinical trial (NCT05020665) [5], paving the way for future MRD biomarker-driven trial designs and accelerated drug approvals.

Persistent somatic mutations in morphologic remission are increasingly used as “MRD” markers but often overlap with those observed in clonal hematopoiesis of indeterminate potential (CHIP) and persist in morphologic remission at high variant allele frequencies (VAF) in the preleukemic clones. Notably, *FLT3*-ITD is increasingly recognized as an important MRD marker, despite its typical late occurrence during leukemogenesis and presence within the leukemic subclone [6]. Whilst the adverse prognostic impact of MRD after intensive chemotherapy and prior to allogeneic hematopoietic cell transplantation (HCT) is well established [7,8], current MRD timepoints, kinetics, and thresholds require further validation in the setting of novel therapies that have become available since 2017 (see recent reviews) [9,10]. Moreover, the role of MRD-directed therapy (i.e., clinical actionability) in AML remains debatable [11,12,13,14], unlike the use of arsenic trioxide in acute promyelocytic leukemia (APL) and blinatumomab in B-cell acute lymphoblastic leukemia (ALL) [15,16].

In this article, we will review (1) the emerging molecular MRD markers, (2) the impact of novel therapeutics on MRD endpoints, and (3) how MRD might be used as a predictive biomarker to guide therapy in AML beyond its prognostic role.

## 2. Emerging MRD Markers

### 2.1. AML-Defining Molecular MRD Markers

Traditionally, MRD assessment aims to measure leukemic blasts below the 5% threshold by the morphologic assessment, detect residual disease, and predict relapse. An ideal molecular MRD marker should be exclusively present within the AML-defining clone, absent (or nearly absent) in the preleukemic state (such as CHIP or myelodysplastic syndrome [MDS]), correlate with therapeutic response, recur at disease relapse, and be technically feasible for molecular tracking. Refer to Table 1 for a summary of the characteristics of AML-defining molecular MRD markers. AML with recurrent genetic abnormalities, as defined by the WHO entities, are the prototypical examples, and some subtypes could define AML regardless of blast count.

The latest WHO classification has expanded the list of recurrent genetic abnormalities that define AML *regardless of blast count*: five specific fusions (*PML::RARA, RUNX1::RUNX1T1*, *CBFB::MYH11*, *DEK::NUP214*, and *RBM15::MRTFA*), three groups of gene rearrangement (*KMT2A*, *MECOM*, and *NUP98*) and one gene mutation (*NPM1*). Whilst these are all potential MRD markers, not all are applied in routine clinical practice due to rare entities, multiple translocation partners (such as *KMT2A*), and rare transcript isoforms (such as non-type A/B/D *NPM1*). It is also worth noting that inv(3), whilst a defining genetic abnormality, is not a gene fusion but rather results in the reposition of a distal *GATA2* enhancer to activate *MECOM* expression. MRD monitoring for t(9;11)/*KMT2A::MLLT3*, the most common fusion among this subgroup, could be particularly helpful to risk further stratification, as this is typically considered intermediate-risk cytogenetics [4,17]. Achievement of MRD negativity for t(9;11) appears to be a pre-requisite for long-term remission and might be useful for early detection of relapse [18]. Data on other *KMT2A::X* fusions also support the utility of MRD monitoring among patients with *KMT2A* fusions, now with increasing relevance with the use of menin inhibitors, which have been shown to induce clinical and MRD responses [19]. Additionally, t(6;9)/*DEK::NUP214*, a high-risk lesion detected in ~1% of patients with AML, is also amenable to serial RT-qPCR to detect persistent or relapsing disease [20].

### 2.2. Persistent Clonal Hematopoiesis with Oncogenic Potential (CHOP)

AML is characterized by a stepwise acquisition of somatic mutations from clonal hematopoiesis to full transformation [21]. At diagnosis, each patient has a median of four to five mutations occurring in >70 genes [22]. Except for *IDH1* Arg132 and *IDH2* Arg140/R172 mutations, the heterogenous molecular landscape most typically requires a targeted next-generation sequencing panel approach (NGS-MRD). A common example observed in de novo AML is the successive occurrence of a mutation in epigenetic regulation (such as *DNMT3A* and *TET2*), followed by an AML-defining mutation (such as *NPM1*), and lastly, a mutation involved in signaling pathways (such as *FLT3*, *NRAS*, and *KRAS*) [6]. These early mutations, also known as preleukemic mutations, overlap with age-related clonal hematopoiesis and often persist in morphologic remission [23]. Whilst sometimes termed “MRD”, the persistence of these mutations is more accurately termed clonal hematopoiesis (CH), and the clinical challenge is to identify CH with oncogenic potential (CHOP) and distinguish this from CHIP (refer to Table 1).

Despite the caveat of persistent mutations overlapping with CHIP, two landmark studies have found that the persistence of mutations, particularly those other than *DTA,* is associated with an increased risk of disease relapse, including those persisting at high VAF consistent with preleukemic origin [24,25]. Other studies evaluating the role of NGS-MRD are summarized in Table 2 [23,24,25,26,27,28,29,30,31,32,33]. In studies comparing both NGS-MRD and MFC-MRD, both modalities are found to be complementary, with decremental outcomes observed for MFC^neg^/NGS^neg^ > MFC^neg^/NGS^pos^ ≈ MFC^pos^/NGS^neg^ > MFC^pos^/NGS^pos^ [23,25,31,32]. The adverse prognostic impact of detectable mutations by NGS might be abrogated by allogeneic HCT in the first remission [25]. Conditioning intensity has been shown to have an impact on the post-HCT outcome of patients with NGS-MRD^pos^ pre-HCT. In patients with no mutations detected, overall survival (OS) did not differ based on conditioning intensity; however, in those with detectable mutations, survival was significantly worse in those who received reduced-intensity conditioning [29,33].

The above findings have led to the ELN recommendation that *DTA* mutations “should be excluded from MRD analysis” [2]. This notion, however, was recently challenged by the ALFA0702 study, including 181 patients (out of 616 achieving CR/CR with incomplete hematologic recovery [CRi]) aged 18–60 years with de novo AML, available DNA sample, and at least one mutation detected at diagnosis. Using an UMI-corrected NGS-MRD panel (TWIST) with a sensitivity of 0.1% VAF, the authors found that persistent *DTA* mutations were associated with a significantly adverse cumulative incidence of relapse (CIR), event-free survival (EFS), and OS. The 4-year CIR in NGS-MRD^neg^ (n = 91, 50%) vs. *DTA* mutations only (n = 42, 23%) vs. NGS-MRD^pos^ (n = 48, 27%) were 68%, 51%, and 39%, and the 4-year OS were 80%, 59%, and 54%, respectively [34]. The potential reasons for this discrepancy from prior studies (such as Jongen-Lavrencic et al. [HOVON-SAKK]) [23] regarding the *DTA* mutations might be related to patient selection (median age 46 years in ALFA0702 vs. 51 years in HOVON-SAKK), treatment protocols, NGS strategies (UMI corrected vs. non-UMI), and assessment timepoints (post-course 1 vs. 2).

Recurrent hotspot mutations affecting *IDH1* Arg132 and *IDH2* Arg140/Arg172 occur in approximately 20% of AML cases, and whilst their relevance as molecular MRD markers remain to be determined, they are of particular importance due to the availability of targeted inhibitors [22,35,36]. Whilst *IDH1/2* mutations are early events in AML, they are rare (0.01%) in CHIP [37,38] and, when detected, were associated with a very high risk (15 out of 15) of subsequent progression to AML [39]. Earlier studies demonstrated that persistent *IDH1/2* mutations after chemotherapy was significantly associated with inferior disease-free survival (3-year disease-free survival 38% vs. 62%) [40] and an increased risk of relapse (59% vs. 24% at 1 year) [41] with no significant impact on OS in both studies. On the contrary, Cappelli et al. studied 150 patients with *NPM1* mutant (mut) AML in remission and suggested that *IDH1*, *IDH2*, *SRSF2*, and *DTA* mutations could be considered together as CHIP-like mutations and did not adversely affect the prognosis. The numbers were small however; 36 patients had mutated *IDH1/2,* of whom only eight had persistence in the mutation [42]. Among patients who had an initial clearance of *IDH1/2*mut clone after therapy, a progressive rise in the *IDH1/2* clone size might portend AML relapse [43]. Although the three hotspot mutations in *IDH1/2* are often considered together, their prognostic impact might differ. In an analysis of patients who received HCT by Bill et al., *IDH2* Arg140 mutations had higher VAF at diagnosis (~50%), lower mutation clearance in morphologic remission, did not have a significant prognostic impact, and thus behaved more like a CHIP-related mutation, while *IDH1* Arg132 and *IDH2* Arg172 were more CHOP-like, including an association of increased risk of relapse [44].

### 2.3. Signaling Pathway Mutations, Focusing on FLT3-ITD

Gene mutations in the signaling pathway, such as *FLT3* (ITD and TKD), *NRAS, KRAS,* and *KIT*, among others, typically occur late in the leukemogenesis and are often subclonal. Hence, whilst detection of these mutations represents residual AML, its absence has a low negative predictive value as AML relapse could occur without these mutations. For example, *FLT3*-ITD could be either lost or gained at the time of AML relapse in 20–30% and 5–10%, respectively [45,46]. Current ELN guidelines recommend that these signaling pathway mutations “are best used in combination with additional MRD markers” [3]. Despite these limitations, several studies have emerged to demonstrate the adverse prognostic impact of *FLT3*-ITD using a highly sensitive NGS-MRD assay, which will be summarized in this section. Other signaling pathway mutations are frequently considered together with other persistent mutations (Section 2.2) and will not be reviewed here, but it is worth noting their role in therapeutic resistance among patients treated with an IDH inhibitor [47], FLT3 inhibitor [48], and venetoclax (VEN) [49].

Conventional fragment analysis by capillary electrophoresis has limited sensitivity in the detection of *FLT3*-ITD at ~1% [50,51]. NGS detection of *FLT3*-ITD has had variable success due to the heterogeneous nature of *FLT3*-ITD mutations, including various insertion sites, insertion lengths (15–300 bp), insertion sequences, >1 mutant clone, and clonal evolution [52,53,54,55,56]. Standard bioinformatics algorithms are not optimized for the detection of larger insertions or deletions (indel). Amplicon-based NGS, with its uniform amplicon start-stop positions, can further complicate indel detection, resulting in discarded unaligned reads. Random fragmentation, such as in Illumina protocols, may break the target ITD sequences and render them undetectable [52,53]. More recently, a highly sensitive, specific, and proprietary *FLT3*-ITD NGS-MRD assay was developed by Invivoscribe^®^ [54]. An open-source bioinformatic pipeline, *getITD*, has also allowed the assay to be more widely implemented in diagnostic laboratories [57].

Using NGS-MRD, we were among the first to demonstrate the adverse prognostic impact of detectable *FLT3*-ITD below the sensitivity of conventional capillary electrophoresis. Among 104 patients with *FLT3*-ITDmut AML undergoing HCT in morphologic CR, 38 (37%) had a detectable *FLT3*-ITD MRD pre-HCT, of which only seven were detectable by conventional capillary electrophoresis. Patients with any *FLT3*-ITD detectable ≥ 0.001% had an inferior 4-year CIR (67% vs. 16%, *p* < 0.001) and OS (≤26% vs. 74%, *p* < 0.0001) [58]. Similarly, Lee et al. identified that 28 (80%) of 35 patients at pre-HCT had *FLT3*-ITD detectable by NGS (sensitivity 0.001%) and identified a threshold of >0.1% to be most predictive of relapse in a multivariate analysis (HR 5.1, *p* = 0.008) [59]. In the pre-MEASURE study (abstract and preprint), using a targeted NGS-based MRD panel (sensitivity 0.01%), *FLT3*-ITD MRD was detectable in 85 of 608 patients (14%) at pre-HCT, which was again significantly associated with risk of relapse and inferior OS [33,60]. In the setting of post-intensive chemotherapy, Grob et al. found that *FLT3*-ITD NGS-MRD was detectable in 47 of 161 (29%) patients after two cycles of intensive chemotherapy and associated with an increased 4-year CIR (75% vs. 33%, *p* < 0.001) and inferior OS (31% vs. 57%, *p* < 0.001) [61]. *FLT3*-ITD NGS-MRD also exceeded the prognostic value of *NPM1* NGS-MRD or MFC-MRD; the data are limited to a direct comparison with the *NPM1* qPCR-MRD assessment [58].

The NGS-based assay also has additional advantages beyond its highly sensitive detection of *FLT3*-ITD. Small *FLT3*-ITD subclones and, thus, clonal heterogeneity are better detected by NGS, although its impact on relapse risk or survival is yet to be determined and is an area that requires further clarity [62,63]. The insertion site can also be determined from NGS. A beneficial effect of midostaurin appeared to be restricted to patients with sole juxtamembrane domain (JMD) insertion sites [64].

## 3. Impact of Novel AML Therapies on MRD Endpoints

After three to four decades of a stagnant therapeutic landscape, we finally saw a series of new drug approvals in AML since 2017 (see recent reviews) [9,10]: FLT3 (midostaurin and gilteritinib) and IDH inhibitors (ivosidenib and enasidenib), VEN, glasdegib, gemtuzumab ozogamicin, CPX-351, and oral azacitidine (AZA). In this section, we will review selected novel therapies with available data on MRD. Table 3 summarizes the impact of novel therapies on MRD endpoints.

### 3.1. VEN and Hypomethylating Agent (HMA)/Low Dose Cytarabine (LDAC)

The use of BCL-2 inhibitor VEN in combination with AZA or low dose cytarabine (LDAC) has changed the treatment landscape in patients with AML ineligible for intensive chemotherapy [65,66]. Improved response rates and OS (median 14.7 months vs. 9.6 months in VEN-AZA vs. AZA alone) from the pivotal phase 3 VIALE-A study led to the FDA approval of VEN-AZA in this population. Despite this, the duration of remission (DOR) was 18 months, and long-term OS was <20% [67]. In 164 patients treated with upfront VEN-AZA evaluable for MFC-MRD, 41% achieved MFC-MRD^neg^ (<0.1%); the rate of MFC-MRD^neg^ was highest (88%) among patients with an *NPM1* mutation [68]. Among patients achieving MFC-MRD^neg^, approximately 50% achieved this by the end of cycle 4, with an additional 27% by the end of cycle 7 and the remainder beyond that. The median DOR and OS were not reached in patients with MFC-MRD^neg^ < 0.1% (12-month estimates were 81.2% and 94.0%), versus the median DOR at 9.7 months and OS at 18.7 months (12-month estimates at 46.6% and 67.9%) in patients with MFC-MRD^pos^ ≥ 0.1%. Other retrospective studies on the prognostic value of MFC-MRD are also consistent with those from the VIALE-A study [69,70].

Molecular MRD data among VEN-treated patients are comparatively more limited. Among those achieving durable remission of >2 years, sustained molecular negativity in *NPM1*mut MRD was observed (n = 4/4 evaluable), whereas molecular clearance of other mutations was variable [49]. In a retrospective analysis of 55 patients who received VEN in combination with AZA (n = 28), decitabine (DEC) (n = 1), or LDAC (n = 26) and who achieved CR/CRi and were evaluable for an *NPM1* qPCR MRD response within the first 6 months, qPCR-MRD^neg^ was achieved in 46%, ≥4 log_10_ reduction in 19%, and <4 log_10_ reduction in 35%. Achievement of qPCR-MRD < 0.005% (per 100 *ABL1*) was associated with a significantly improved OS (88% vs. 34% at 18 months) at a median follow-up time of 24.3 months, updated from <0.2% in the initially published abstract [71].

Other novel VEN-based combinations outside of AZA, DEC, or LDAC have resulted in high response rates, but longer follow-up is required to ascertain the durability of remission. VEN-cladribine-LDAC alternating with VEN-AZA was assessed in 60 patients with newly diagnosed AML unfit for intensive chemotherapy; the response rate was 93%, and MFC-MRD^neg^ (<0.1%) was achieved in 84% of 51 patients with sample availability, resulting in a significant difference in 2-year OS compared with patients with MFC-MRD^pos^ (80% vs. 45%, HR 3.97, *p* = 0.016) [72]. In attempts to improve the induction regimen, fludarabine, cytarabine, idarubicin, and granulocyte colony-stimulating factor (FLAG-IDA) combined with VEN in the newly diagnosed setting resulted in an 89% composite CR rate, of which 93% attained MFC-MRD^neg^ (<0.1%) with demonstrable survival benefit (median OS NR vs. 16 months, *p* = 0.03) [73].

In the future, it is conceivable to apply MRD monitoring in the VEN-based low-intensity cohort to consider cessation of therapy in a select group of patients who achieve MRD negativity [74,75].

### 3.2. FLT3 Inhibitors

*FLT3*-ITD MRD response after the use of FLT3 inhibitors is of particular interest. The improved OS, despite similar protocol-specified CR in the RATIFY study, has been attributed to deeper MRD remission from the addition of midostaurin, but this has not been verified [76,77]. Among patients treated with midostaurin and intensive chemotherapy in the single-arm phase 2 AMLSG16-10 trial, significantly lower CIR (HR 0.1, *p* < 0.001) and favorable OS (HR 0.27, *p* < 0.03) were observed for 87% of patients who achieved *FLT3*-ITD NGS-MRD^neg^ at the end of treatment [78].

The phase 3 ADMIRAL study led to the FDA approval of gilteritinib as the treatment for relapsed/refractory (R/R) *FLT3*mut (ITD and TKD) AML, where 34% of patients achieved CR or CR with partial hematologic recovery (CRh) compared with 15% in the conventional salvage chemotherapy group with improved OS (median 9.3 months vs. 5.6 months) [79]. Considering the *FLT3*-ITD NGS-MRD data from the phase 1/2 CHRYSALIS study [80], gilteritinib resulted in *FLT3*-ITD NGS-MRD^neg^ (<0.01%) in 8 (16%) of 49 patients who achieved CR/CRi, which was associated with a longer OS (median 131.4 weeks vs. 43.3 weeks, *p* = 0.066). To further improve the response in R/R *FLT3*-mutated AML, the combination of VEN and gilteritinib was recently studied in 56 patients, of which 75% achieved a composite morphologic remission (including CR/CRi/CRp/MLFS): 15 (60%), 11 (44%) and 5 (20%) of 25 evaluable achieved an NGS-MRD < 1%, <0.1%, and 0.01%, respectively. However, OS among those with an *FLT3*-ITD NGS-MRD of <1% was not significantly different (median OS 11.6 months vs. 8.2 months, *p* = 0.16). More data are needed to conclusively determine the impact of *FLT3*-ITD NGS-MRD clearance on the outcomes of patients with R/R *FLT3*-mutated AML [81].

Preliminary data from the QuANTUM-First study showed that the upfront addition of quizartinib to intensive chemotherapy in patients with an *FLT3*-ITD mutant AML resulted in improved OS with placebo (median 31.9 months vs. 15.1 months, *p* = 0.03) and a lower post-induction *FLT3*-ITD NGS-MRD level (median VAF 0.01% vs. 0.03%, *p* = 0.02). *FLT3*-ITD MRD level < 0.01% post-induction, which was observed in 24.6% of quizartinib and 21.4% of placebo-treated patients, was associated with a longer OS (median OS NR vs. 29.4 months) irrespective of treatment arm. Interestingly, attaining an undetectable MRD level (<0.001%), which was observed in a higher proportion of patients on quizartinib (13.8% vs. 7.4%), did not seem to confer a significant survival benefit [82].

These studies highlight the feasibility and importance of the prospective incorporation of a highly sensitive *FLT3*-ITD NGS-MRD assessment in clinical trials. Further studies are required to further inform its kinetics and prognostic utility, particularly in the setting of FLT3 inhibitors and R/R AML, and to predict those with emerging therapeutic resistance when treated with other novel therapeutics.

### 3.3. IDH1 and IDH2 Inhibitors

Ivosidenib (IDH1 inhibitor) and enasidenib (IDH2 inhibitor) were both FDA-approved for R/R *IDH*-mutated AML based on the respective single-arm monotherapy studies [83,84,85]. Additionally, ivosidenib was also approved for newly diagnosed AML in unfit patients [83,86]. Ivosidenib monotherapy in R/R AML resulted in CR/CRh rates of 30%, of which 21% achieved *IDH1*-MRD clearance, associated with improved DOR (median 11.1 months vs. 6.5 months) and survival (median OS 14.5 months vs. 10.2 months) [83]. In the newly diagnosed setting and treatment with ivosidenib monotherapy, CR/CRh rate was 42.4%, of which the rate of *IDH1*-MRD^neg^ was 64% [86]. Similarly, enasidenib monotherapy in R/R AML resulted in an overall response rate of 38.8% (including MLFS and partial remission), of which 12% achieved *IDH2*-MRD^neg^, associated with a significant survival benefit (median OS 22.9 months vs. 8.8 months, *p* = 0.0153) [85].

IDH inhibitors have also been studied in combination studies. In newly diagnosed patients with AML unfit for intensive chemotherapy (phase 3 AGILE study), ivosidenib and AZA resulted in higher rates of *IDH1*-MRD^neg^ as compared with ivosidenib and a placebo (17/33 [51.5%] vs. 3/10 [30.0%]) [87]. All baseline co-mutations were also cleared among 73% of patients who achieved CR [88]. Similarly, an enasidenib–AZA combination in unfit newly diagnosed AML (phase 1b/2 AG221-AML-005) resulted in greater *IDH2* mutation reduction; the median VAF was 0.023% in CR vs. 0.233% in CRi/CRp/PR/MLFS vs. 0.872% in no response [89]. Combining ivosidenib with VEN ± AZA in both newly diagnosed and R/R patients in a phase 1b/2 study resulted in *IDH1*-MRD^neg^ and MFC-MRD^neg^ in 67% and 63% of patients, respectively, resulting in significant improvement in survival (median OS NR vs. 8 months, *p* = 0.002) [90]. In newly diagnosed AML, ivosidenib or enasidenib in combination with intensive chemotherapy resulted in *IDH1* and *IDH2*-MRD^neg^ of 16/41 (39%) and 15/64 (23%) and MFC-MRD^neg^ in 16/20 (80%) and 10/16 (63%) among patients achieving CR/CRi/CRp, respectively [91].

### 3.4. Gemtuzumab Ozogamicin (GO)

GO is a CD33 antibody–drug conjugate that has demonstrated survival benefits when added to standard intensive chemotherapy in patients with non-adverse karyotype AML [92,93]. AML with mutated *NPM1* is one of the AML subtypes with the highest CD33 expression, which has been associated with the efficacy of GO [94]. Molecular MRD responses among patients with *NPM1*mut AML receiving upfront GO have been examined by the ALFA-0701 (n = 77) and AMLSG 09-09 trials (n = 469 evaluable out of 588) [95,96]. In the ALFA-0701 study, patients aged 50–70 years were randomized to 7 + 3 induction chemotherapies ± fractionated doses of GO, followed by two consolidation cycles with daunorubicin and cytarabine ± GO. The proportion of patients achieving *NPM1* qPCR-MRD < 0.1% was significantly higher in patients treated with GO, both post-induction (39% vs. 7%, *p* = 0.006) and at the end of treatment (91% vs. 61%, *p* = 0.028) [95].

In the AMLSG 09-09 trial, patients received two cycles of induction therapy with ATRA, idarubicin, cytarabine, and etoposide, followed by up to three consolidation cycles of high dose cytarabine and ATRA. Patients randomized to GO received this on day 1 of two induction cycles and the first consolidation cycle; significantly lower *NPM1* qPCR-MRD levels were found in BM and PB post-induction and maintained throughout subsequent treatment cycles. This was also reflected by a significantly lower 4-year CIR in patients who received GO (31.6% vs. 43.9%, *p* = 0.015) and a superior relapse-free survival (RFS) (60.5% vs. 48.9%; *p* = 0.028) [96].

In a smaller cohort of patients with core-binding factor (CBF) AML on the UK MRC AML15 trial, the addition of GO increased molecular MRD reduction after only one course of induction in patients with t(8;21), but this was not mirrored in the inv(16) cohort and with no perceptible effect on relapse or survival [97].

### 3.5. CC-486

CC-486, an oral AZA formulation, was granted FDA approval in 2020 as maintenance therapy in patients who are unable to complete intensive chemotherapy after achieving remission post-intensive induction, based on OS benefit in the phase 3 QUAZAR study [98]. This benefit was seen irrespective of the presence of MFC-MRD^pos^ (≥0.1%) in 46% of patients at study entry. On serial assessment, CC-486 resulted in an increased rate of MFC-MRD^neg^ (<0.1%) compared with placebo at 37% vs. 19%, with 76% of patients achieving this within 6 months of treatment [99]. Achievement of MFC-MRD^neg^ (including 60 patients on CC-486 or placebo) in the study was associated with longer survival: median OS 41.3 months vs. 9.0 months and median RFS 20.4 months vs. 2.8 months (MRD-MFC^neg^ vs. remained MFC-MRD^pos^). Post-hoc analyses of the *NPM1*mut subgroup showed that CC-486 was able to eradicate flow MRD (<0.1%) in 63% of patients vs. 33% in placebo (*p* = 0.051) and with longer duration of sustained MFC-MRD^neg^ compared with placebo (15.6 months vs. 7.1 months; *p* = 0.006). Similarly, patients who were *FLT3*mut had a numerically higher rate of MRD conversion to MFC-MRD^neg^ (50% in CC-486 vs. 18% in the placebo arm, *p*-value not provided) [100].

**Table 3 ijms-24-04790-t003:** Impact of novel therapies on MRD endpoints.

Therapy	Population	N	MRD Marker	Thresholds (% of N)	Timepoint	Outcomes (neg vs. pos) or (< vs. ≥ Threshold)
**Venetoclax (VEN)**						
VEN-AZA [68]	New AML	164	MFC	BM < 0.1% (41)	Any	Median OS NR vs. 19 m 1 y OS 94% vs. 68%
VEN-DEC10 [69]	New AML	83	MFC	BM < 0.1% (54)	1–4 m	Median OS 25 m vs. 7 m (2 m timepoint)
VEN-AZA [70]	New AML	63	MFC	BM < 0.1% (38)	1–3 m	18 m CIR 13% vs. 57% 18 m OS 70% vs. 35% Median OS 26.5 m vs. 10 m
VEN-HMA/LDAC [71]	New AML	55	*NPM1* cDNA	BM < 0.005% (49)	≤6 m	18 m OS 87% vs. 39%
VEN-CLAD-LDAC alternating with VEN-AZA [72]	New AML	51	MFC	BM < 0.1% (84)	Any	2 y OS 80% vs. 45% Median DFS NR vs. 5.9 m
VEN-FLAG-IDA [73]	New AML	40	MFC	BM < 0.1% (93)	Any	Median OS NR vs. 16 m
**FLT3 inhibitor**						
Quizartinib [82]	New AML	161	*FLT3*-ITD	BM < 0.01% (24.6)	PC1	Median OS NR vs. 29.4 m
Gilteritinib [80]	R/R	49	*FLT3*-ITD	BM < 0.01% (16)	Any	Median OS 131.4 weeks vs. 43.3 weeks
VEN-Gilteritinib [81]	R/R	25	*FLT3*-ITD	BM < 0.1% (60; 20% < 0.01%)	Any	Median OS 11.6 m vs. 8.2 m
**IDH1/2 inhibitors**						
Ivosidenib + intensive chemotherapy [91]	New AML	41	*IDH1*	BM < 0.02–0.04% (39)	Any	Not reported
		20	MFC	BM < 0.1% (80)	Any	Not reported
Ivosidenib [83]	R/R	34	*IDH1*	BM < 0.02–0.04% (21)	Any	Median OS 11.1 m vs. 6.5 m
Ivosidenib [86]	New AML	14	*IDH1*	BM < 0.02–0.04% (64)	Any	Not reported
Ivosidenib + VEN ± AZA [90]	New AML or R/R	31	*IDH1*	BM < 0.1–0.25% (67)	Any	Median OS NR vs. 8 m
			MFC	BM < 0.1% (63)		
Enasidenib + intensive chemotherapy [91]	New AML	64	*IDH2*	BM < 0.02–0.04% (23)	Any	Not reported
Enasidenib [85]	R/R	101	*IDH2*	BM < 0.02–0.04% (12)	Any	Median OS 22.9 m vs. 8.8 m
**Gemtuzumab ozogamicin**						
ALFA-0701 [95]	New AML	61	*NPM1* cDNA	BM < 0.1% (25)	PC1	2 y CIR 21% vs. 55%
				BM < 0.1% (78)	EOT	2 y CIR 45% vs. 67%
AMLSG 09-09 [96]	New AML	232	*NPM1* cDNA	BM reduction ≥ 3 log_10_ (87)	PC2	4 y CIR 28.5% vs. 60%
				PB neg (53)	PC2	4 y CIR 18% vs. 53%
				PB neg (78)	EOT	4 y CIR 28% vs. 70%
**CC-486**						
CC-486 [99]	AML in first remission	236	MFC	BM < 0.1% (37)	Any	Median OS 41.3 m vs. 9.0 m Median RFS 20.4 m vs. 2.8 m

Abbreviations: AZA, azacitidine; BM bone marrow; CLAD, cladribine; CIR, cumulative incidence of relapse; DEC10, ten-day decitabine; DFS, disease-free survival; ddPCR, digital droplet polymerase chain reaction; EOT, end of treatment; FLAG-IDA, fludarabine, cytarabine, idarubicin, and filgrastim; LDAC, low dose cytarabine; m, months; MFC, multiparameter flow cytometry; MRD, measurable residual disease; neg, negative; NR, not reached; OS, overall survival; PB, peripheral blood; PC1, post course 1; PC2, post course 2; pos, positive; RT-qPCR, reverse transcription quantitative polymerase chain reaction; RFS, relapse-free survival; and R/R, relapsed/refractory.

## 4. MRD-Guided Therapy in AML

MRD assessment improves risk stratification in AML beyond the baseline patient and disease characteristics [101,102], but the key question remains “does it merely portend fate”? In other words, is MRD a predictive biomarker for therapeutic targeting that translates to an improvement in RFS and/or OS after accounting for lead time bias when compared with intensive salvage chemotherapy (followed by HCT) in patients with frank hematologic relapse? The benefit of relapse prevention needs to be balanced with treatment toxicities. An ideal MRD marker (Table 1) will reliably predict relapse in ensuing weeks/months, allowing time for intervention (including donor preparation), in addition to being prognostic regarding an increased risk of relapse over the next months to years. In parallel, a clinician will be readier to pre-emptively treat an emerging relapse if the therapy has fewer toxicities (see Section 3 on the novel therapies). The use of MRD to guide HCT in CR1, particularly in patients with favorable and intermediate-risk AML, has been reviewed elsewhere: patient selection vs. deference, donor selection, conditioning intensity/regimen, and post-HCT strategies, including immunosuppressant, disease monitoring, and therapeutic intervention [103,104,105,106]. Despite an earlier retrospective study by Araki et al. showing similarly poor outcomes post-myeloablative HCT between those with MFC-MRD^pos^ morphologic remission and active morphologic disease [107], more recent studies, including both retrospective (European Society for Blood and Marrow Transplantation registry) [108] and prospective randomized-controlled trials (BMT-CTN 0901 study) [29], demonstrated improved outcomes following myeloablative conditioning HCT in those with pre-HCT MRD positivity of various markers.

This section will review the MRD-guided non-HCT therapeutic approach (summarized in Table 4), noting that the strategy of MRD eradication/reduction pre-HCT (vs. direct HCT) is unproven but rapidly evolving.

### 4.1. MRD-Directed Therapy Using AZA

One of the earliest descriptions of MRD-directed therapy (excluding acute promyelocytic leukemia) was the use of AZA in 10 patients with *NPM1* qPCR-MRD relapse or persistence > 1%, of which seven (70%) had an MRD response (≥1 log_10_ reduction) and remained in morphologic remission at a median follow up of 10 months (range 2–12) [109]. This concept was extended to the prospective phase 2 RELAZA2 study, where 53 patients with falling post-HCT donor chimerism < 80% or MRD transcript levels > 1% at any time post-chemotherapy or HCT were pre-emptively treated with AZA. Sixty percent of patients were *NPM1*mut, and overall, 31 (58%) patients had an MRD response, including 19 (36%) who achieved MRD^neg^. The study met its primary endpoint, with 58% of patients remaining relapse-free at 6 months from therapy initiation. Overall, the 2-year RFS was 46% [14].

The value of MRD-guided therapeutic decision-making was further highlighted in a retrospective analysis by Short et al., where 55 patients with MFC-MRD relapse either continued current therapy (n = 36; including three with no further therapy) vs. those who had a change of therapy to direct HCT (n = 9) or HMA-based treatment (n = 7). Survival outcomes were significantly better among those who had a change in therapy; the 5-year RFS was 31% vs. 5% (*p* = 0.01), and OS was 45% vs. 17% (*p* = 0.01) [111]. Among the seven patients who received HMA-based treatment, three achieved MRD negativity (followed by HCT in 2), one remained MRD positive (then HCT), and three relapsed.

### 4.2. MRD-Directed Therapy Using Intensive Chemotherapy

The CETLAM group performed *NPM1* qPCR-MRD monitoring on 110 patients with ELN favorable *NPM1*mut AML in first remission (CR1) after standard chemotherapy, of which 33 patients experienced molecular failure, defined as failure to achieve *NPM1*-MRD ≤ 0.05% after consolidation therapy (n = 11) or MRD relapse (n = 22). MRD-directed therapy was at the discretion of the treating clinician: 13 direct HCT, 12 salvage chemotherapy or HMA (10 followed by HCT), and 8 had interim morphologic relapse [110]. Eighty percent of patients receiving MRD-directed intensive chemotherapy achieved negative MRD. HCT was realized in 70% and 52% of patients with molecular failure and morphologic relapse, respectively. A survival benefit was observed in patients treated with molecular failure with 2-year OS 86% vs. 42% (*p* = 0.0014). With the limitation of small numbers, no significant survival difference was observed between patients who received pre-transplant MRD-directed therapy compared with direct HCT. Important caveats include that the *NPM1* qPCR-MRD threshold of 0.05% at the end of consolidation is considered MRD detectable at a low level and might not be associated with increased relapse risk [2], and *NPM1* persistence after intensive chemotherapy has been shown to spontaneously resolve or persist at a low level in ~39% of patients [115].

In the NCRI AML17 trial, intensive chemotherapy (usually FLAG-IDA) was given to 27 patients with molecular relapse, of whom 16 (59%) achieved MRD negativity prior to HCT. This approach was at the discretion of the clinician. A comparison with HCT was not possible as only three other patients with molecular relapse directly proceeded to HCT without attempted MRD eradication [112].

### 4.3. MRD-Directed Therapy Using VEN

Several studies have applied VEN as MRD-directed therapy, predominantly in the *NPM1*mut cohort, which is recognized as a predictive biomarker of response to VEN [49]. In a small cohort of seven patients treated with VEN and AZA or LDAC for *NPM1* qPCR-MRD relapse after prior intensive chemotherapy, MRD^neg^ was achieved in 86% of patients within 1–2 cycles, with ongoing molecular remission at a median time of 10.8 months follow-up [11]. We further studied this in the prospective phase 2 VALDAC study, utilizing VEN-LDAC in 48 patients with either MRD relapse (n = 26; 20 were *NPM1*mut) or early morphologic relapse (n = 22). MRD response (≥1 log_10_ reduction) was achieved in 69% (after a median of 1 cycle), including 54% achieving MRD^neg^ (after a median of 2 cycles), resulting in an estimated 2-year EFS of 55% and OS of 73% [12].

In a UK-wide program reporting on outcomes of VEN-based low-intensity combinations (AZA, LDAC, another agent, or monotherapy), 19 patients were treated with MRD failure (18 MRD relapse and one persistent MRD), with 84% achieving molecular remission. In an indirect comparison with 103 patients treated in parallel with a morphologic disease, patients treated with molecular MRD failure demonstrated superior OS (median 18.4 months vs. 7.1 months, *p* = 0.004) [113].

### 4.4. MRD-Directed Therapy Using FLT3 Inhibitors

The use of FLT3 inhibitors (gilteritinib, quizartinib, or sorafenib) was described in 48 patients with known *FLT3*-ITD at baseline, who experienced a molecular MRD failure: *NPM1* (81%), *NUP98::NSD1* (8%), *DEK::NUP214* (6%), and one of each *CBFB::MYH11* and *RUNX1::RUNX1T1* (4%). MRD failure was defined as persistence > 2% (n = 6) or MRD relapse (n = 42). In 47 patients evaluable for response, MRD response was achieved in 64%, and 40% achieved MRD^neg^. Fifty percent of patients received subsequent HCT. Two-year OS and EFS were 80% and 73%, respectively [114].

### 4.5. Future Directions in MRD-Directed Therapy

The future direction in MRD-directed therapy comprises integrated and coordinated frameworks to ascertain, firstly, the clinical actionability of various markers and urgently assess the survival benefit of treating MRD failure to circumvent protracted research timelines, high cost of singular clinical trials, and high attrition rates. Two large clinical trials and initiatives with the goal of harmonizing MRD monitoring and assessing the role of pre-emptive therapy are currently underway. The AMLM26 INTERCEPT study (ACTRN12621000439842) is a platform trial in Australia and the US, has been actively recruiting since August 2022, and aims to examine novel therapies as MRD-directed intervention in patients in first/second remission with MRD response as a primary endpoint, and key secondary endpoints of RFS, time to and duration of MRD response and OS [116]. Refer to Figure 1 for the AMLM26 INTERCEPT trial schema. This trial is linked to a coordinated MRD monitoring framework guided by an MRD reference committee once patients enter the platform in first/second remission. If suspected MRD failure occurs, this is confirmed by centralized laboratories, and patients are then allocated to one of many biomarker-directed treatment arms; the same is applied to patients where morphologic failure eventuates during MRD monitoring. For example, a patient with *NPM1*mut MRD relapse will be allocated to VEN-LDAC, or if with *FLT3*-ITDmut MRD relapse, to VEN-gilteritinib. Close clonal tracking using molecular assays ± MFC-MRD or leukemic stem cell flow MRD employed to assess response and clonal evolution is incorporated. Patients who experience MRD or morphologic relapse can rotate to another biomarker-driven treatment arm in the study. The MyeloMATCH precision medicine clinical trial (NCT05564390) is an umbrella trial comprising four tiers, which aims to enroll patients with newly diagnosed AML/MDS, who are then followed throughout their treatment journey with planned activation in mid-2023 [117,118]. The highest tier (tier 4) is aimed at targeting MRD, with incorporated technologies such as MFC and duplex sequencing.

With the increasing use of novel agents in the MRD failure setting, it is imperative that ongoing efforts are focused on the identification of clonal evolution, recognition of treatment-emergent resistance mechanisms, and unraveling the intrinsic and extrinsic biological differences between leukemia at the stage of MRD vs. morphologic failure.

## 5. Conclusions

At this stage, emerging molecular MRD markers have not consistently yielded a high positive predictive value for relapse or oncogenic potential, and beyond technical optimization, which focuses on achieving a higher sensitivity and lower limit of detection, ongoing work will need to dissect the genetic heterogeneity of AML at single-cell level and assess if the putative MRD clone resides within the leukemic compartment in the MRD setting [119,120]. These efforts are currently prohibited by the high cost of multiomic technologies and technical complexity. The ultimate goal is that the right assay for the right mutation in the right scenario and at the right timepoint is used.

Our article has also shown that novel therapies have the capacity to induce high rates of MRD response in patients attaining morphologic remission when used in the bulk disease setting, with exploratory analyses demonstrating a survival benefit in patients attaining MRD negativity. Moving forward, the prognostic value of historical MRD thresholds at specific timepoints will require prospective revalidation in the setting of novel therapeutics, as many of the aforementioned studies are limited by small numbers for MRD analyses.

Finally, MRD-directed therapy in AML is now the mainstay of two large, harmonized, biomarker-driven trial initiatives aimed at the efficient evaluation of pre-emptive therapy using a multi-modal approach. Novel therapies have shown promise in inducing high MRD response rates and with the possibility of lower toxicity. Whilst HCT remains the gold standard for potential cure in AML, the area needing urgent resolution is the value of pre-HCT MRD-directed therapy in patients with pre-HCT MRD detection vs. direct to HCT and whether or not this will alter post-HCT fate. This will hopefully provide an answer to the age-old question of whether MRD-directed intervention in AML is just delaying the inevitable or if it will ultimately improve patients’ long-term outcomes.

## Figures and Tables

**Figure 1 ijms-24-04790-f001:**
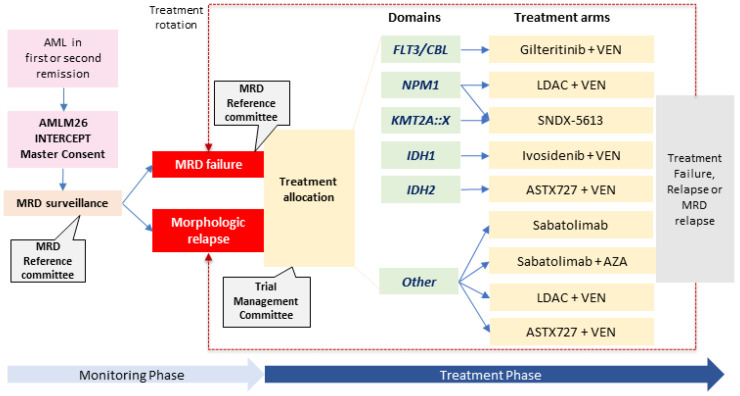
Trial schema of the AMLM26 INTERCEPT for MRD-directed therapy in AML. Abbreviations: AML, acute myeloid leukemia; AZA, azacitidine; LDAC, low dose cytarabine; MRD, measurable residual disease; and VEN, venetoclax.

**Table 1 ijms-24-04790-t001:** Established and emerging molecular markers according to mutation type.

	Mutation Types	Clonal Hematopoiesis	AML-Defining	Signaling Pathway
Characteristics				
Examples		*DNMT3A*, *TET2*, *ASXL1*, *IDH1*, *IDH2*, *TP53*, *SRSF2*, and *RUNX1*, among others	*NPM1*, *CBFB::MYH11*, *RUNX1::RUNX1T1*, *PML::RARA*, and *KMT2A* fusions, among others	*FLT3* (ITD and TKD), *NRAS*, *KRAS*, *KIT*, *CBL*, and *PTPN11*, among others
Clonal hierarchy		Preleukemic, but overlaps with leukemia-initiating clone	Leukemia-initiating clone	Subclone
Presence in clonal hematopoiesis		Yes, but some mutations are deemed higher risk than others (“high risk CCUS”)	Rare	Rare
Clearance post morphological remission		Frequently persists	Yes	Yes
Predictive value		Low value for *DTA* mutations (see text), and some likely have more oncogenic potential	High if detectable above the threshold or if serially increasing (MRD relapse)	High positive predictive value but low negative predictive value
Stable at relapse		Yes	Yes, but note branched evolution	No
Sample source		gDNA	Usually cDNA	gDNA
Molecular assays		Targeted NGS panel ddPCR (only for hotspots)	RT-qPCR (or RT-dPCR) NGS-MRD panel (gDNA)	Targeted NGS panel *FLT3*-ITD NGS-MRD panel

Abbreviations: CCUS, clonal cytopenias of undetermined significance; cDNA, complementary DNA; ddPCR, digital droplet PCR; *DTA*, *DNMTA, TET2*, and *ASXL1*; gDNA, genomic DNA; ITD, internal tandem duplication; MRD, measurable residual disease; NGS, next-generation sequencing; PCR, polymerase chain reaction; RT-qPCR, reverse transcription quantitative PCR; RT-dPCR, reverse transcription digital PCR; TKD, tyrosine kinase domain.

**Table 2 ijms-24-04790-t002:** Summary of studies evaluating MRD by targeted NGS panel in AML.

Study	N	PB vs. BM	Timepoint	Mutations	Thresholds (% of N)	Risks (neg vs. pos) (or < vs. > Threshold))
Klco, 2015 [24]	50	BM	PC1	Any	<2.5% (52)	Median OS 42.2 m vs. 10.5 m
Morita, 2018 [25]	122	BM	PC1	Any ^a^	Negative (48)	2 y CIR 24% vs. 46% 2 y OS 77% vs. 60%
Jongen-Lavrencic, 2018 [23]	430	Either	PC2	Non-*DTA*	Negative (72)	4 y CIR 31.9% vs. 55.4% 4 y RFS 58.1% vs. 36.6% 4 y OS 66.1% vs. 41.9%
Rothenberg-Thurley, 2018 [26]	126	Either	First remission	Any ^b^	<2% (60)	Median RFS 55.7 m vs. 11.7 m Median OS NR vs. 31.3 m
Thol, 2018 [27]	96	Either	Pre-HCT	Non-*DNMT3A*	Negative (55)	5 y CIR 17% vs. 66% 5 y RFS 74% vs. 31% 5 y OS 78% vs. 41%
Kim, 2018 [28]	104	BM	Post-HCT D21	Any	<2% (85)	3 y CIR 16.0% vs. 56.2% 3 y OS 67.0% vs. 36.5%
Hourigan, 2020 [29]	190	PB	Pre-HCT	Any ^c^	Negative (35)	Neg: 3 y OS 56% vs. 63% (MAC vs. RIC) Pos: 3 y OS 61% vs. 43% (MAC vs. RIC) Pos: 3 y CIR 19% vs. 67% (MAC vs. RIC)
Heuser, 2021 [30]	131	Either	Post-HCT D90 ± 180	Non-*DTA*	Negative (80)	5 y CIR 25% vs. 62% 5 y RFS 68% vs. 35% 5 y OS 73% vs. 49%
Patkar, 2021 [31]	201	BM	PC1	Any	Negative (29)	3 y CIR 25.7% vs. 47.5% Median RFS NR vs. 17 m Median OS NR vs. 27 m
Tsai, 2021 [32]	335	BM	PC2	Non-*DTA*	Negative (71)	Median CIR 4.8 y—NR vs. 0.6–1.1 y Median OS NR vs. 3.1–3.6 y
Hourigan, 2022 (Pre-MEASURE) [33]	1075	PB	Pre-HCT	*FLT3*, *NPM1*, *IDH1*, *IDH2*, and *KIT*	<0.01% (70)	3 y CIR 23% vs. 62% 3 y RFS 59% vs. 25% 3 y OS 66% vs. 36%

Abbreviations: BM, bone marrow; CIR, cumulative incidence of relapse; EOT, end of treatment; HCT, hematopoietic cell transplantation; m, month; MAC, myeloablative conditioning; n.s., nonsignificant; neg, negative; NR, not reached; OS, overall survival; PB, peripheral blood; PC1, post-course 1; pos, positive; RFS, relapse-free survival; RIC: reduced-intensity conditioning; and y, year. ^a^ Additional analysis showed a stronger prognostic association when *DTA* was removed from the analysis. However, a separate analysis was not performed to compare the outcomes among those with *DTA* persistence vs. clearance. ^b^ Separate analysis showed no prognostic significance of *DNMT3A* mutation persistence. ^c^ Includes only 13 genes: *DNMT3A*, *TET2*, *ASXL1*, *TP53*, *RUNX1*, *NRAS*, *KIT*, *IDH1*, *JAK2*, *SF3B1*, *IDH2*, *FLT3*, and *NPM1.*

**Table 4 ijms-24-04790-t004:** Summary of MRD-guided therapy in AML.

Study	Population	N	Treatment	MRD Response	Survival
Sockel, 2011 [109]	*NPM1* MRD relapse or persistence > 1%	10	AZA	≥1 log_10_ decrease (70%)	Not reported
Platzbecker, 2018 [14]	RT-qPCR > 1% or donor chimerism loss	53 (32 *NPM1*mut)	AZA	36% MRD^neg^	1 y RFS 46%
Bataller, 2020 [110]	*NPM1* MRD failure (ELN 2017 favorable risk)	33 (Eight morphologic relapses)	20 chemo/HMA ± HCT 13 direct HCT	80% MRD^neg^ (8/10 chemo)	2 y OS 86%
Short, 2022 [111]	MFC-MRD relapse	16	Seven HMA-based chemo Nine direct HCT	43% MRD^neg^ (3/7 chemo)	5 y RFS 31% 5 y OS 45%
Dillon, 2020 [112]	*NPM1* MRD relapse	30	27 chemo + HCT Three direct HCT	59% MRD^neg^ (16/27 chemo)	2 y OS 63%
Tiong, 2021 [11]	*NPM1* MRD relapse	Seven	VEN + HMA/LDAC	86% CR MRD^neg^	Not reported
Wood, 2022 [113]	Molecular MRD failure (marker not specified)	19 (103 morphologic disease)	VEN ± LDAC or HMA or other	84% molecular remission	Median OS 18.4 m
Tiong, 2022 [12]	MRD relapse	26 (20 *NPM1*mut)	VEN-LDAC	≥1 log_10_ decrease (69%) 54% MRD^neg^	2 y EFS 54% 2 y OS 73%
Othman, 2022 [114]	MRD (*NPM1* or other gene fusions) failure with baseline *FLT3*mut	48 (39 *NPM1*mut)	32 gilteritinib Eight quizaritinib Eight sorafenib	40% MRD^neg^	2 y OS 80%

Abbreviations: AZA, azacitidine; chemo, chemotherapy; EFS, event-free survival; HCT, hematopoietic cell transplant; HMA, hypomethylating agent; LDAC, low dose cytarabine; m, months; MRD, measurable residual disease; mut, mutant; neg, negative; OS, overall survival; RFS, relapse-free survival; RT-qPCR, reverse transcription-quantitative polymerase chain reaction; VEN, venetoclax; and y, year.

## Data Availability

Not applicable.

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
