# Peer review of "Targeting Measurable Residual Disease (MRD) in Acute Myeloid Leukemia (AML): Moving beyond Prognostication"

_ijms, 2023, doi:10.3390/ijms24054790_

Round 1

Reviewer 1 Report

The strength of this review is that it is provides an up to date (including abstracts from ASH 2022) overview of progress in molecular MRD assays, MRD measured efficacy with associated outcomes for the more novel therapies and some data for interventions directed by MRD failure.  The tables are comprehensive and helpful summaries of recent studies.  The authors should however recognise that  there is already evidence  that MRD does not ‘merely portend fate’ if patients  can be allografted (from the trials of  CR1 MRD directed HCT and the CTN study showing the effect of myeloablative conditioning).  Also, section 5 should highlight  that conclusions from many of the studies commented on in sections 3 and 4 are limited by very small numbers in the MRD analyses.  

Minor comments are detailed below.

There are some issues with the syntax at the start of this review.

Lines 48-49 <On the contrary, FLT3-ITD is also increasingly recognized as an important MRD marker>.  This sentence needs rewording to better link with previous sentence. < on the contrary> to what? -obviously not persistent somatic mutation mutations since Flt3 is a somatic mutation and may be persistent.

Line 52 Change sentence to 'require further validation  in the setting of  the novel therapies that have  become available from 2017

Line 57-58 <will review 1) the emerging molecular MRD markers and contrast these with the traditional validated MRD markers>.  There is not much ‘contrasting’ going on in this review. Do the authors mean traditional validated molecularMRD markers as there is almost nothing relating to flow cytometric MRD assays?  Could reword this simply as 1) developments in molecular MRD markers.

Line 69. <An ideal MRD marker …. should be cleared after chemotherapy>.  Not necessarily if the chemotherapy or other treatment is ineffective,  the MRD  marker needs to be a measure of response.

Line 97, <Except for  IDH mutations>  - insert also  FLT3 TKD mutations.

Description of ASH 22 presentation of NGS MRD in ALFA0702 study would benefit from a comment as to potential reasons for  discrepancy for example age and treatment . 

Line 138 <whilst their relevance as molecular MRD markers remains to be dated>, presumably typo for ‘dated’

Line 140 typo for <event> , should be <events>.

Line 232 <qPCR-MRD <0.005% (per 100 ABL1)> -should this not be <0.2%?

Section 3.5 should also include comment with reference https://doi.org/10.1182/blood.2022016293

Reviewer 2 Report

The evaluation of treatment efficiency in  AML, especially in the form of MRD measurement, represents one of the extremely important aspects, considering its role in evaluating the prognosis and  indication for a stem cell transplantation. A promising therapeutic direction of treatment is represented by the series of new agents recently approved in AML therapy that may represent MRD-driven treatment options.

The authors present a useful review of molecular markers evaluable  for MRD assessment, the impact of recently approved therapies on MRD status, and clinical trials studying the value of MRD guided therapy in AML.

I recommend publication in present form

Author Response

Thank you for the positive feedback.